graph theory

complex networks, resilience, robustness, trophic coherence, rich-core club

**Author for correspondence:**
Alessio Pagani
e-mail: apagani@turing.ac.uk

# Resilience or robustness: identifying topological vulnerabilities in rail networks

Alessio Pagani[1], Guillem Mosquera[1,2], Aseel Alturki[3], Samuel Johnson[5], Stephen Jarvis[3], Alan Wilson[1], Weisi Guo[1,4] and Liz Varga[6]

[1]The Alan Turing Institute, London, UK
[2]Mathematics Institute, [3]Department of Computer Science, and [4]School of Engineering, University of Warwick, Coventry, UK
[5]School of Mathematics, University of Birmingham, Birmingham, UK
[6]School of Management, Cranfield University, Cranfield, UK

AP, 0000-0002-8404-8890; WG, 0000-0003-3524-3953

Many critical infrastructure systems have network structures and are under stress. Despite their national importance, the complexity of large-scale transport networks means that we do not fully understand their vulnerabilities to cascade failures. The research conducted through this paper examines the interdependent rail networks in Greater London and surrounding commuter area. We focus on the morning commuter hours, where the system is under the most demand stress. There is increasing evidence that the topological shape of the network plays an important role in dynamic cascades. Here, we examine whether the different topological measures of resilience (stability) or robustness (failure) are more appropriate for understanding poor railway performance. The results show that resilience, not robustness, has a strong correlation with the consumer experience statistics. Our results are a way of describing the complexity of cascade dynamics on networks without the involvement of detailed agent-based models, showing that cascade effects are more responsible for poor performance than failures. The network science analysis hints at pathways towards making the network structure more resilient by reducing feedback loops.

## 1. Introduction

Cascade delays and cancellations on rail transport can cause devastating economic damage and dent consumer satisfaction. Existing knowledge either focuses on improving operational

practices or considers a pure topological analysis. We advance this by considering both real passenger travel flows and the network topology together. This creates a stronger understanding of its dynamic vulnerability and resilience. In earlier years, research largely focused on improving specific functionalities in rail systems; and more recent research has focused on the relationship between the general network topology and whether this has macroscopic bearing on the overall system performance [1]. The efficiency of transport networks has been related with their resilience [2] and the different types of topologies have been analysed, comparing the network geometry and the level of connectivity. However, these studies predominantly focus on the pure topological characteristics of a graph [3,4].

## 1.1. Related work

### 1.1.1. UK rail network

The UK rail network transports more than 1.7 billion passengers per year, of which 1.1 billion passengers commute in and around London.[1] According to the Office of Rail and Road,[2] last year in and around London, only 86.9% of passenger trains arrived on time and 4.8% of the journeys were cancelled or significantly late. Often these delays are interrelated and the relationship between cascade effects and network dynamics is not well understood.

In the current literature, most of the proposed studies consider natural or man-made disasters, but they do not consider the stress of the network during the peak-hours and how the structure of the network created by the massive flows of people can influence their ability to maintain a good service. For example, several graph-based approaches have been proposed to improve the performances by revising the design and maintenance of the rail networks [5],[3] but do not consider dynamic passenger flows. Other studies focus on specific extreme scenarios [6][4] or unfavourable conditions [7] that cause disruptions.

As our data show, under the same external conditions, the major rail companies in and around London show dramatically different performance levels. In this work, we **hypothesize** that this difference can, in part, be attributed to the peak passenger demand. A coupling relationship between flow and network structure can tease out the indicative measures that correlate strongly with overall performance.

### 1.1.2. Vulnerability of transport networks

The concept of vulnerability of transportation network, introduced in the literature by Berdica [8], is generally defined as the susceptibility to disruptions that could cause considerable reductions in network service or the ability to use a particular network link or route at a given time. Many have applied general network science disruption analysis. For example, several studies [9–11] have been conducted for modelling railway vulnerability with promising predictive results. Bababeik *et al.* [12] recently proposed a mathematical programming model that is able to identify critical links with consideration of supply and demand interactions under different disruption scenarios. Recent work has also used graph properties to infer interaction strengths and use an epidemic spreading model to predict delays in railway networks [13].

## 1.2. Innovation

In this paper, we take a systems-of-systems approach by applying a complex network analysis to transport networks. Unlike prior studies that focus only on the topological aspects of the network, we consider several important additional aspects which attempt to match our analysis to reality. First, we consider passenger volumes during morning commuter or rush-hour, which weights the network and adds directionality. The morning rush-hour is important because most of the delays and the highest economic impact of delays occur during this time. Second, we consider a multiplex of different urban overground, regional and national rail services (both together and separately). As a result, we have a weighted and directed multiplex network, which requires more sophisticated network analysis methods to uncover its resilience

---

[1]Passenger rail usage. Office of Rail and Road. See http://dataportal.orr.gov.uk/browsereports/12. (10 September 2018).

[2]Passenger and freight rail performance. Office of Rail and Road. See http://dataportal.orr.gov.uk/browsereports/3. (10 September 2018).

[3]2015 *Smart, green and integrated transport*, European Commission, Horizon 2020, Work Programme 2016–2017.

[4]*Railway specifications? The specification and demonstration of reliability, availability, maintainability and safety (RAMS)*. Systems Approach to Safety. European Committee for Electrotechnical Standardization. BS EN 50126-2:2017.

and robustness to cascade failures. Finally, we map our network resilience and robustness results to actual railway performance figures of delay and cancellation statistics and consumer satisfaction.

## 1.3. Analysis

Vulnerability is a major problem in the study of complex networks and it can be regarded as the susceptibility of a networked system to suffer important changes in its structure and dynamic functions under any form of disruption. When such disruptions affect the internal state of the nodes (e.g. stations) or links (e.g. train lines) of the network, it becomes important to predict the extent of such perturbations under the perspective of dynamical systems (e.g. linear stability analysis); throughout this paper, we refer to this problem as the study of **resilience**. Resilience is important for understanding cascade effects that suppress the performance of the network, such as cascade delays due to signal failures or poor scheduling. In plain terms, resilience describes the problem of a train from A to B that is late, which will affect the ensuing service B back to A using the same train. But, when the perturbations involve some sort of attack or out-right failure (e.g. a disruption in a station due to someone walking on to the tracks or a signal failure), the challenge tends to be in studying the resulting connectivity loss and secondary loss of functionality in neighbouring stations. We refer to this as the **robustness** problem, which is very different from the aforementioned resilience. In plain terms, robustness considers when a train from A to B will be halted if the track in between is blocked or station B is closed.

The concepts of resilience and robustness on networks admit various interpretations and definitions [14,15]. A generally accepted definition of stability is applicable when the system performance returns to a desirable state. For homogeneous linear stability, one might equate resilience with equilibrium points and look at the leading eigenvalue of the Jacobian matrix [16]. When linear stability is not suitable due to complex dynamics, many authors [17–21] have studied system resilience from different perspectives. Some consider the dynamic response (e.g. time to recovery) of the whole system after a specific disruption [21], while others use random perturbations to numerically quantify system response [22].

In terms of robustness, a common definition is that the number of nodes that must be removed in order for the network to break down is a popular measure of its robustness [23]. While such approaches depend strongly on assumptions about the system, they generally map railway systems well [24].

However, such approaches depend strongly on assumptions about the system, such as details of the dynamics or the number of neighbours required for a node to function. In this work, we use them instead of recent advances in the ecological system analysis to study resilience and robustness, which can be obtained directly from the adjacency matrix (even for weighted and directed networks) and have been found to be good proxies for resilience and robustness in ecosystems (figure 1). While there are certainly differences between ecosystems and rail systems, both are essentially transport networks in which either biomass or passengers flow from sources (plants or home towns) through various intermediary nodes, and end in sinks (top predators or work places).

### 1.3.1. Resilience of weighted and directed networks

We introduce a parameter for quantifying the resilience of weighted directed networks measuring their *trophic coherence*. Trophic coherence is a property of directed graphs that defines how much a graph is hierarchically structured. The rationale is that hierarchical systems have fewer feedback loops and are less likely to suffer from cascade effects.

When networks are modelled as a discrete linear time invariant (LTI) system with a defined input and output [25], the dynamic response stability is defined by the location of roots of its transfer function (negative domain). In such a case, the absence of feedback loops ensures stability. The presence of feedback loops will cause non-zero roots and risk instability. When we consider a complex network with $\sim N^2$ input output combinations, the transfer function cannot be defined. As such, we measure the overall network incoherence, which is a compressed figure of merit for how many feedback loops exist [26,27]. Johnson *et al.* [26,27] proved that 'a maximally coherent network with constant interaction strengths will always be linearly stable', and that it is a better statistical predictor of linear stability than size or complexity. We measured the coherence of the network through the *incoherence parameter*, a measure of how tightly the trophic distance associated with edges is concentrated around its mean value (which is always 1) [26].

To define trophic coherence in a directed network, the first step is to define basal nodes (i.e. nodes that predominantly supply energy—high out-degree and low in-degree). That is to say, stations with a high trophic level *receive* passengers while stations with a low trophic level *provide* passengers. Thus, basal nodes are likely to be home train stations of commuters.

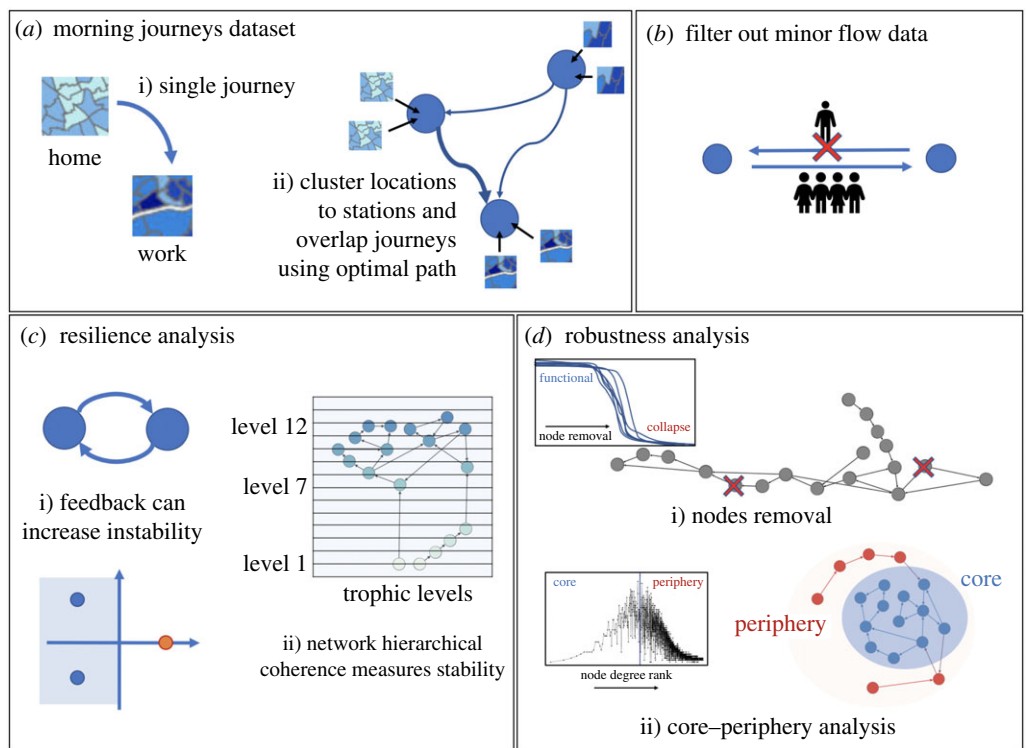

**Figure 1.** We reconstruct the major rail networks under stress conditions considering the morning journeys (*a*) and we measure the topological characteristics of these networks: the uninteresting flows are removed (*b*), then the resilience (*c*) and robustness (*d*) of these networks are analysed.

Defining trophic coherence in real data networks requires some pre-processing: unlike the already studied networks in other works (e.g. food webs [28,29]), the London urban rail network in peak-hours does not have predefined basal nodes (i.e. nodes with in-degree of 0). In transportation, this means that there is always a non-zero passenger counter-flow travelling from urban to the countryside stations during the morning rush hour. To distil the basal nodes from the data, we developed and tested two different approaches (we show these techniques in §4) to identify the basal nodes in networks where they do not naturally exist. In other words, many weighted networks do not have apparent energy sources. In the first proposed approach, we apply **basal node enforcement**, whereby basal nodes are selected based on their network centrality characteristics (e.g. out-degree). The trophic level of the remaining nodes is then computed using the standard formula (equation (4.2)). In the second proposed approach, we apply **passenger flow filtering**, which identifies the basal nodes by reducing the connectivity of the network. We sequentially reduce the connectivity (by increasing a flow threshold) until basal nodes emerge naturally.

### 1.3.2. Robustness

The objective in this case is to use both proxy and direct measures of robustness. Direct measures are random or targeted node removal. However, as robustness is not rigorously defined, proxy measures may yield a more holistic insight: a variety of robustness measures are used to compare against real railway performance. As such, we are using a variety of robustness measures to establish a wider evidence base.

Firstly, as a proxy, we evaluate its *core* and *periphery* meso-scale structure. The core periphery ratio gives a scalable and compressed understanding of robustness, and the argument is formalized by Borgatti *et al.* [30]. Another proxy measure is the *rich-club* coefficient [28,29,31,32]. Secondly, we evaluated the robustness directly by performing sequential node removal [33]. The nodes of the rail networks are removed randomly and the network connectivity is then studied, evaluating the size of the largest strongly connected component.

### 1.4. Data

In this study, we analysed the rail network under demand stress conditions (morning rush-hour). The commuter paths are computed considering the information relative to places where people live and

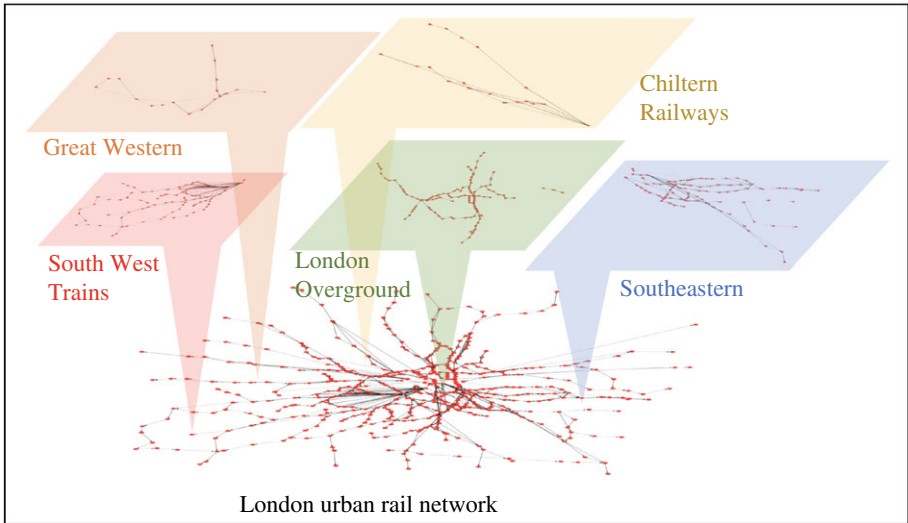

**Figure 2.** Directed graph of passenger flows during morning peak-hours.

work provided by the UK National Census Transformation Programme.[5] The optimal travel paths were provided by the National Rail (including rail services through underground tunnels, but not including the underground/subway system) through their *TransportApi* service.[6] Given an origin station and a destination station, the *TransportApi* service provides all the information about the travel, including the intermediate stop stations. We first check if rail travel is required for a person to go from home to work, and if so, we compute their optimal journey and use these data to weight the network. In the current study, only the travels that start and end in a bounding area of 80 km from central London have been taken into account (this approximately covers Cambridge to the north, Oxford to the northwest, Reading to the west and Brighton to the south). It roughly represents all 1 h commuter paths, which is the national standard according to ONS.[7]

The resulting dataset represents the flows of people in morning peak-hours on the rail network (available on Dryad), when they travel from their homes to their places of work. Each journey is defined as a set of two or more stations (in case of intermediate stops of the train all the intermediate stations are included). The dataset is transformed in a *directed weighted graph* where the nodes are the train stations, the edges are the weighted flows of passengers and a journey is an ordered set of nodes that includes the departure station, the arrival station and any intermediate station (if the train stops, as we consider the service class of the train).

A **directed graph** is defined [34] as an ordered pair $G = (N, E)$, where $N$ is a set of nodes (i.e. stations) and $E$ is a set of ordered pairs of nodes, called edges (i.e. trains that go from a node to the following). When, in our graph, one or more passengers are going from node $i$ to node $j$ (or these two nodes are intermediate stations of the travel), an edge $e_{ij}$ is added to the graph. The weight of this edge is the sum of all the passengers of the journeys that include travels from node $i$ to node $j$. The directed graph of the passenger flows during morning peak-hours is shown in figure 2. We show the whole multiplexed network, as well as some examples of the individual sub-networks comprising urban overground (London Overground), regional links (Thameslink) and national services (e.g. Southern rail).

## 2. Results

The delays in a rail network and, more in general, the performances of the service are influenced by the topological structure of the network. The notion is that a more resilient and/or robust network should guarantee lower cascade delays and faster recovery in case of disruptions. In this chapter, the different multi-scale rail networks (this includes both local London overground rail services and national major

[5]*Census transformation programme*, Office for National Statistics. See https://www.ons.gov.uk/census/censustransformation programme. (May 2018).

[6]*TransportApi*, National Rail. See https://www.transportapi.com. (May 2018).

[7]2018 *Two-hour daily commute 'for millions'*, BBC News. (online) See http://www.bbc.co.uk/news/uk-38026625. (25 May 2018).

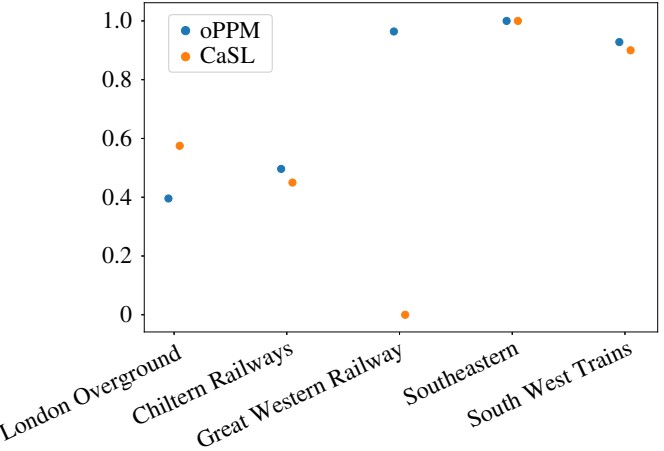

**Figure 3.** oPPM versus CaSL correlation.

**Table 1.** Number of nodes (stations) per company in the morning peak-hours network.

| name | nodes |
| --- | --- |
| London Overground | 109 |
| Great Western Railway | 18 |
| Chiltern Railways | 18 |
| South West Trains | 91 |
| Southeastern | 64 |

rail services) in morning peak-hours are analysed separately and the results are compared with the **Public Performance Measures** provided by the ORR (Office of Rail and Road),[8] an independent regulator that monitors the rail industry's health and safety performance. ORR holds Network Rail[9] the company that, with 20 000 miles of track, owns, operates and develops Britain's railway infrastructure.

In particular, two performance measures are used in our comparison:

**PPM**. The *Public Performance Measure* combines figures for punctuality and reliability into a single performance measure. Usually, it shows the percentage of trains which arrive at their terminating station within 5 min (for London and South East and regional services) or 10 min (for long distance services).[10] In this paper, for the sake of clarity, we define the **oPPM** with the opposite value of the PPM (oPPM = 100% − PPM). oPPM is the percentage of trains which do not arrive at their terminating station within 5 or 10 min (depending on the distance).

**CaSL**. The *Cancellation and Significant Lateness* is a percentage measure of scheduled passenger trains which are either cancelled (including those cancelled en route) or arrive at their scheduled destination more than 30 min late (see footnote 10).

The performance measures used in this paper are referred to the year 2017 (key statistics by train operating company (TOC)—2016–2017[11]). To provide statistically significant results (small networks are more sensitive to local functional effects than macroscopic topological structure), we considered the five companies with the highest number of nodes in the network, excluding companies with very simple network structures (e.g. Heathrow Express has only one line). The companies taken into account and the number of stations are shown in table 1.

Four out of five analysed rail companies (figure 3) show a strong correlation between these two measures, while in one case (Great Western Railway) these values are not correlated, possibly

[8]*Office of Rail and Road - who we are*, Office of Rail and Road. See http://orr.gov.uk/about-orr/who-we-are. (May 2018).

[9]*Network Rail - about us*, National Rail. See https://www.networkrail.co.uk/who-we-are/about-us/. (May 2018).

[10]*Public performance measure*, National Rail. See https://www.networkrail.co.uk/who-we-are/how-we-work/performance/public-performance-measure/. (May 2018).

[11]*Statistical releases*, Office of Rail and Road. See http://orr.gov.uk/statistics/published-stats/statistical-releases. (May 2018).

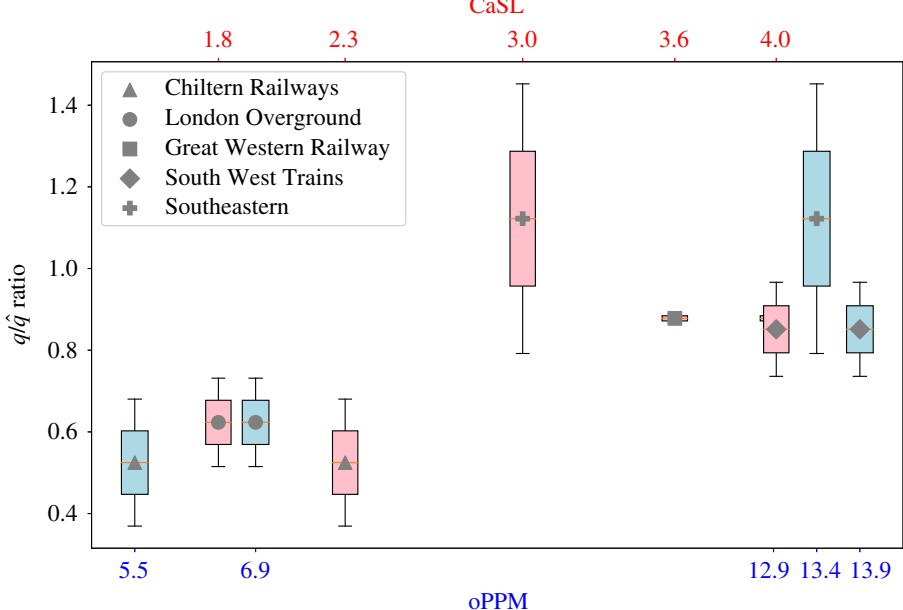

**Figure 4.** oPPM and CaSL compared with trophic incoherence parameter of each rail company.

meaning that this company often has little delays (low resilience) but generally does not have major disruptions (high robustness).

The Pearson correlation coefficient (PCC) [35] is used to establish if there is a correlation between the topology parameters of the network and the performance measures. PCC has a value between $+1$ and $-1$, where 1 is total positive linear correlation, 0 is no linear correlation and $-1$ is total negative linear correlation. Two variables with a correlation coefficient greater than 0.7 are considered *highly correlated*, while they are considered *moderately correlated* when the PCC coefficient is between 0.3 and 0.7.

## 2.1. Trophic incoherence analysis

The degree to which the rail networks are coherent (or incoherent) can be investigated by comparison with a null model. We use the basal ensemble expectation $\tilde{q}$ as a null model to compare the incoherence parameter of our rail networks. The trophic incoherence measure $q/\tilde{q}$ has a value close to 1 when a network has a trophic coherence similar to a random expectation, it has a value lower than 1 when the network is coherent while it has a value greater than 1 when the network is incoherent (more details and the computation of the basal ensemble expectation are provided in §4.1.1).

The incoherence coefficient $q$ of a morning peak-hour network is computed using the *passenger flow filter* method with different flow filtering thresholds, between 1 and 4, with a granularity of 0.5 (details on the selection of the methodology and the parameters are given in §4.1). The average of the computed incoherence parameters is compared with the relative oPPM and the CaSL measures and shown in the following figures along with their standard deviation.

Our results exhibit a highly positive correlation between the trophic incoherence of the network and the Public Performance Measure (PCC = 0.98), suggesting that there is a high correlation between the resilience of a rail network and the probability of its trains to arrive at their terminating station on time. There is also a high positive correlation between the trophic incoherence and the Cancellation and Significant Lateness measure (PCC = 0.92), evidencing also a correlation between low resilience and the percentage of trains either cancelled or that arrive to their destination with more than 30 min late. The trophic incoherence measure $q/\tilde{q}$ compared with oPPM and CasL is shown in detail in figure 4: more coherent networks (low $q/\tilde{q}$) are generally associated with lower delays (oPPM) and cancellations (CaSL).

## 2.2. Rich-club coefficient analysis

The rich-club coefficient of a network is an indicator of the robustness of a network, in particular we say that networks with a rich-club coefficient greater that 1 are characterized by the rich-club phenomenon. The rich-club coefficient measures how many nodes of degree at least $k$ are connected in a graph (how

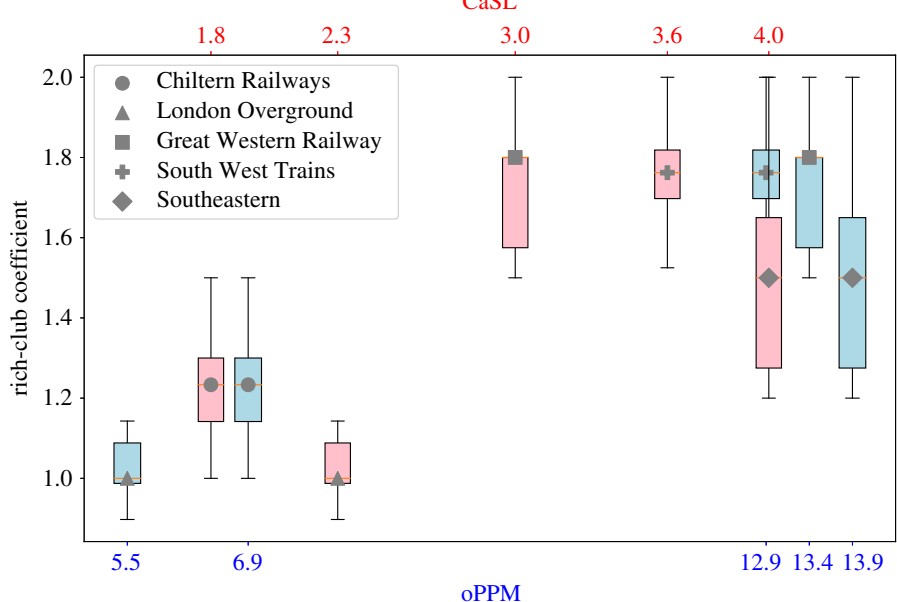

**Figure 5.** oPPM and CaSL compared with rich-club coefficient of each rail company.

many edges are present), normalized by the maximum number of possible connections between them (maximum number of edges) in a complete graph (details are provided in §4.2.1). In this paper, we used the standard definition of rich-core, classifying the nodes according to their degree.

We compared the highest coefficient reached (considering all the possible $k$ values) for each company with its performances metrics (figure 5). Our results show that even if there is a moderate correlation between the value of the rich-club coefficient and the performances (PPM has PCC = 0.62 and CaSL has PCC = 0.55), there is no evidence of any correlation between the presence of the rich-club phenomenon and the service performances of the companies.

## 2.3. Core size analysis

The size of the core of a network compared to its periphery represents the percentage number of well-connected core stations, versus the sparse periphery stations (as we discussed, intuitively a network with a bigger core has more connections between the stations and, thus, a higher robustness to disruptions). In this section, we compare the percentage of core nodes of each company network, computed using *degree* and *trophic* level for ranking, and its oPPM and CaSL measures.

Our findings show that there is a moderate positive correlation between the size of the degree core (PCC = 0.38) and the trophic core (PCC = 0.59) of a company network and the oPPM. Instead, there is no correlation with the CaSL (degree core PCC = −0.09, trophic core PCC = 0.28). A comparison between the degree and trophic core of the companies and the performance measures is shown in figures 6 and 7.

## 2.4. Removal of random node analysis

We attacked the network removing nodes and analysing the size of the largest component. The experiments were repeated several times removing the nodes randomly: in figure 8 is shown the average size of the largest component of each company and its standard deviation. The threshold value, $T = 50\%$ (dashed red line in the figure), has been chosen as the limit value to consider a network 'alive', the percentage of nodes required to destroy a network (connectivity < 50%) is defined as its *robustness to attacks*. The robustness to attacks is compared with the performance measures of the companies (figure 9). Our results show a strong correlation between the robustness to attacks and the CaSL measure (PCC = 0.83) and a moderate correlation (PCC = 0.58) with the oPPM measure.

The correlation analysis has been extended to other significant company statistics (figure 10) showing that oPPM and CaSL are correlated with some of the other values, but the incoherence ratio $q/\tilde{q}$ is indeed

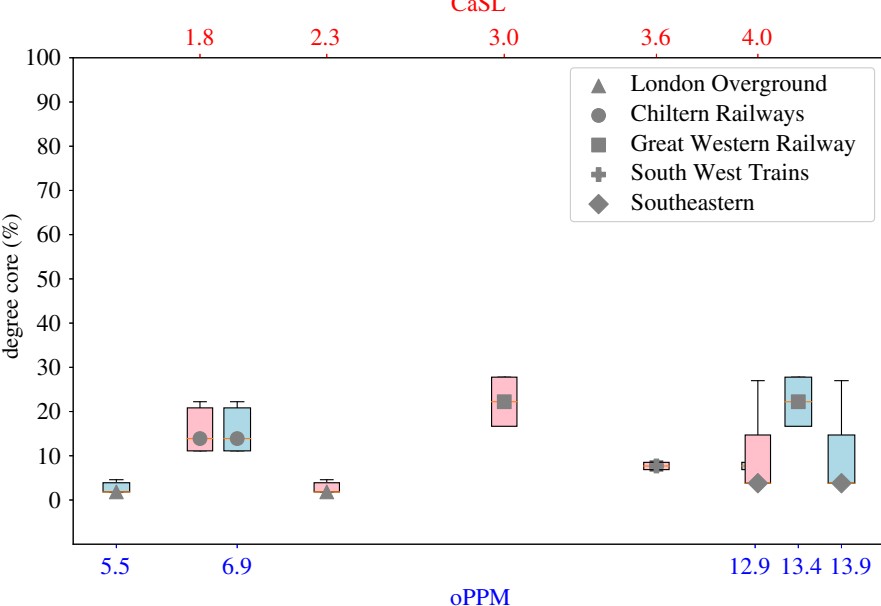

**Figure 6.** oPPM and CaSL compared with degree core of each rail company.

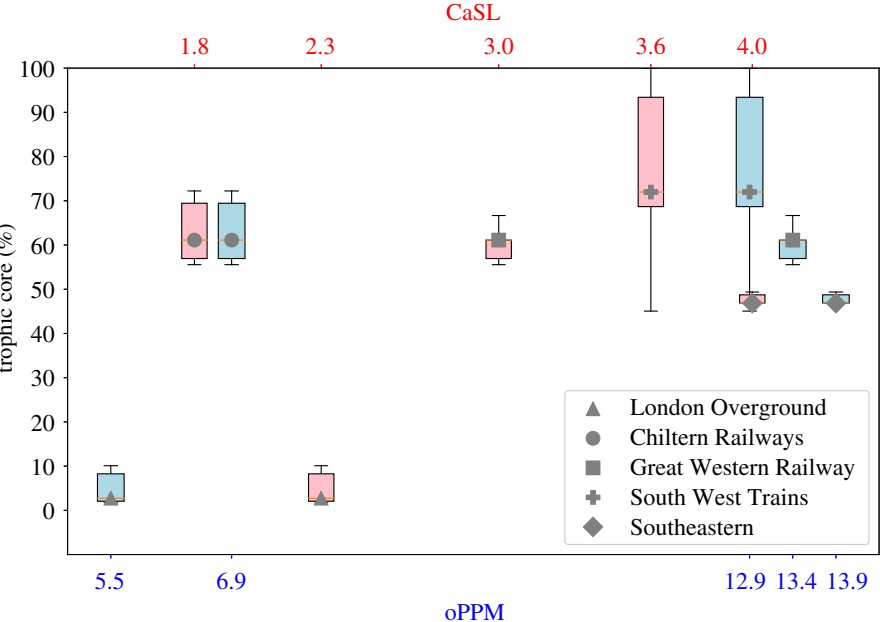

**Figure 7.** oPPM and CaSL compared with the trophic core of each rail company.

one of the most significant. The robustness to attacks has shown to be a good indicator for cancellations and significant delays (CaSL). The size of the core (both degree and trophic cores) and the rich-club phenomenon do not provide any significant correlation with performances. The size of the rail network in terms of the number of employees and stations also has a strong correlation to oPPM, which is likely to be the effect that larger networks are more likely to have feedback loops and incur cascade effects.

# 3. Discussion

In this work, we proposed a study of London's urban rail network under stress conditions, during the morning peak-hours. We represented the major companies' rail networks as weighted directed graphs, where the nodes indicate the stations, the edges indicate the flows of people and the weights of the

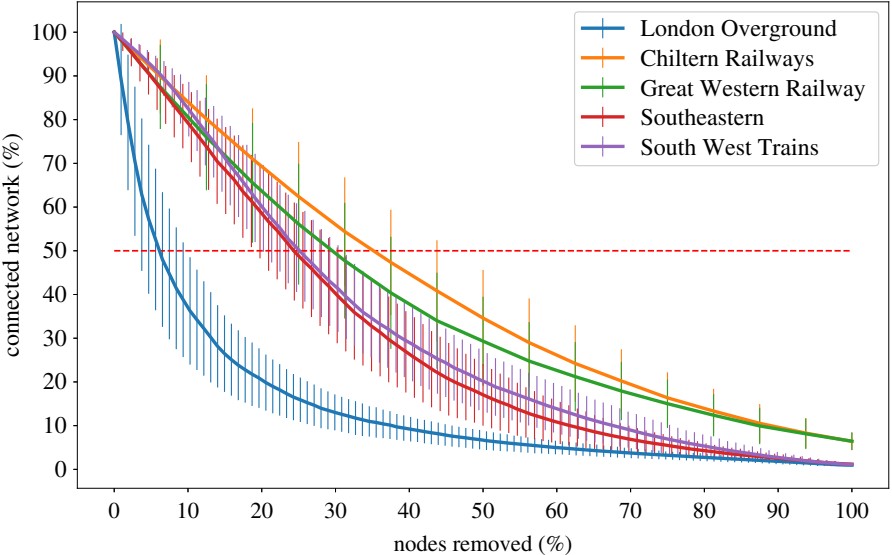

**Figure 8.** Size of the largest strongly connected component. The horizontal line indicates when more than 50% of the network is compromised and it is used as the value to compute the robustness in figure 9.

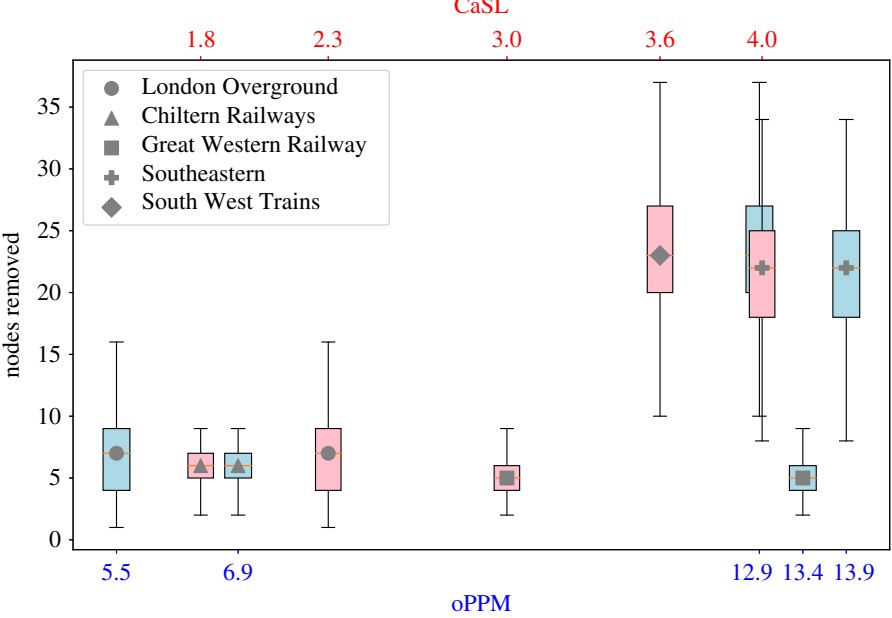

**Figure 9.** oPPM and CaSL compared with the robustness to attacks.

edges the number of people travelling on that segment. If two stations are connected but there are no passengers travelling in the morning, these stations are considered disconnected (there is not an edge between these nodes).

We studied the resilience and the robustness of these networks drawing inspiration from techniques used to study natural complex networks, such as food webs. Our results suggest that the resilience indicators ($q$ and $q/\tilde{q}$) have a strong correlation with the performance parameters PPM (Public Performance Measure) and CaSL (Cancellations and Significant Lateness). Conversely, the different robustness indicators (size of the core and rich-club phenomenon) are not significantly correlated with these measures, although the robustness to attacks is correlated with the CaSL measure.

There is further interesting research that the community can act using our data and building on our methods. First, an interesting improvement in the current work would be the introduction of a theoretical model that could help assess the role of noise—understood as deviations from shortest

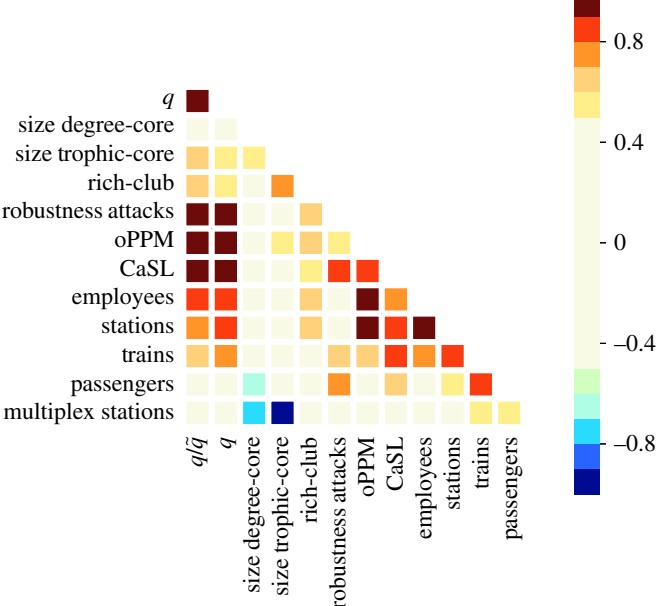

**Figure 10.** Pearson correlation coefficient between different measures and indicators. Incoherence parameter/basal ensemble ($q/\tilde{q}$), incoherence parameter ($q$), size of degree core (size degree-core), size of trophic core (size trophic-core), rich-club phenomenon (rich-club), robustness to attacks (attacks), 100%—public performance measure (oPPM), cancellations and significant lateness (CaSL), number of employees, number of stations, number of trains, number of passengers and multiplex stations (stations that connect different companies).

path routing protocols between origin and destination [36]—in the design of resilient flows in complex networks. In particular, we hypothesize the existence of a trade-off between resilience and travel distance mediated by the amount of noise present in the network flows: strictly shortest path protocols will tend to overload links when the network is attacked [37,38], an effect that could be mitigated by adding randomness in the rerouting. Intuitively, artificially inducing random behaviour (noise or dynamic routing with randomness) in human behaviour may even alleviate overall congestion and thus achieve a lower global travel time. Second, we have in this paper compressed complex network dynamics into a simple coherence metric. Given that we want to maximize coherence, the reverse open question is how best to make the minimum number of changes in scheduling and/or graph to maximize a positive step change in coherence. This problem is open ended, because there are clearly local economic reasons for loops in transport and only a limited number of train carriages to serve them.

# 4. Methods

## 4.1. Trophic coherence and resilience

The trophic level of a node $i$, called $s_i$, is defined as the average trophic level of its in-neighbours, plus 1:

$$s_i = 1 + \frac{1}{k_i^{\text{in}}} \sum_j a_{ij} s_j, \tag{4.1}$$

where $a_{ij}$ is the adjacency matrix of the graph and $k_i^{\text{in}} = \sum_j a_{ij}$ is the number of in-neighbours (in degree) of the node $i$. Basal nodes $k_i^{\text{in}} = 0$ have trophic level $s_i = 1$ by convention.

By solving the system of equations (4.1), it is always possible to assign a unique trophic level to each node as long as there is at least one basal node, and every node is on a directed path which includes a basal node [26]. In our study, the trophic level of a station is the average level of all the stations from which it receives passengers plus 1. For this reason, stations near residential areas in the suburbs will have lower trophic level than those close to business areas in the centre.

Each edge has an associated *trophic difference*: $x_{ij} = s_i - s_j$. The distribution of trophic differences, $p(x)$, always has mean 1, and the more a network is trophically coherent, the smaller the variance of this

distribution. We can measure trophic coherence with the *incoherence parameter q*, which is simply the standard deviation of $p(x)$:

$$q = \sqrt{\frac{1}{L}\sum_{ij} a_{ij} x_{ij}^2 - 1},$$ (4.2)

where $L = \sum_{ij} a_{ij}$ is the number of connections (edges) between the stations (nodes) in the network. A perfectly coherent network will have $q = 0$, while a $q$ greater than 0 indicates less coherent networks.

In our study of the morning peak-hour rail networks, there are not natural basal nodes. To solve the equations for the trophic levels computation, we defined two methodologies to identify them: the *basal nodes enforcement* and the *flows filtering*.

### 4.1.1. Comparison with null model

The degree to which empirical networks are trophically coherent (or incoherent) can be investigated by comparison with a null model. The **basal ensemble expectation** $\tilde{q}$ can be considered a good approximation for finite random networks [27]. We use this parameter as a null model to compare the incoherence parameter of our empirical networks.

The basal ensemble expectation for the incoherence parameter is [27]:

$$\tilde{q} = \sqrt{\frac{L}{L_b} - 1},$$ (4.3)

where $L$ = number of edges in the network. $L_b$ = number of edges connected to basal nodes.

The ratio $q/\tilde{q}$ is used to analyse the coherence of the network: a value close to 1 shows a network with a trophic coherence similar to a random expectation. Values lower than 1 reveal coherent networks, while values greater than 1 reveal incoherent ones.

Johnson & Jones [27] found that food webs are significantly coherent ($q/\tilde{q} = 0.44 \pm 0.17$), metabolic networks are significantly incoherent ($q/\tilde{q} = 1.81 \pm 0.11$) and gene regulatory networks are close to the random expectation ($q/\tilde{q} = 0.99 \pm 0.05$).

### 4.1.2. Basal nodes enforcement

The first technique used to select the basal nodes revolves around the enforcement of the desired number of basal nodes, selecting them according to some properties of the nodes. This technique enforces a predefined number *EN* of nodes to be basal nodes (their trophic level is imposed 1). The nodes to be enforced are selected according to their similarity to real basal nodes, namely the nodes with the lowest ratio between incoming and outgoing edges. More formally, the $k^{out}/k^{in}$ ratio is computed for all the nodes, then the trophic level of the *EN* nodes with the lower ratio is enforced to 1 ($s_i = 1$). If parts of the network are not connected to basal nodes, only the largest strongly connected component is considered. This technique maintains the structure of the network intact (it does not add/remove nodes or edges) but, instead, it does not take into account its natural topology when selecting the basal nodes, making the selection artificial: the selection of the number of basal nodes is artificially defined by the user and does not evaluate the ideal natural number of basal nodes present in the network.

### 4.1.3. Flows filtering

In the analysis of the morning peak-hour commute, the factors that determine the stability of the network depend on the major flows of people (from home to work commute). The paths with just a small portion of commuters can thus be ignored. To remove these paths, a threshold $T$ for the detection of the major flows is defined: when two nodes $i$ and $j$ are connected with two edges $a_{ij}$ and $a_{ji}$, the edges whose ratio $a_{ij}/a_{ji}$ is below the threshold $T$ are deleted.

With this approach, it is possible to remove those loops in the network that are not relevant for the peak-hour analysis (e.g. 100 people going from node $i$ to $j$ and only 1 going from $j$ to $i$, the edge $(j, i)$ can be removed without degrading the quality of the peak-hour flows study). If $T \geq 1$, for each pair of nodes, only the edge with the highest weight is maintained, and only if it is greater than the other (otherwise both the edges are removed). The larger the $T$, the more the direction of the flow from one station another has to be predominant compared to the reverse. If $T < 1$, both edges could be possibly maintained if their flows were balanced (the lower the $T$, the more unbalanced the flows).

With this technique, the basal nodes are not enforced but rather naturally emerge from the change in the structure of the network (i.e. the edges with a low impact on the study are removed from the network). The higher the threshold, the more edges are removed. However, the threshold has to be accurately chosen because a high threshold could lead to the removal of interesting flows, reducing the information in the graph and providing incomplete results.

## 4.2. Core – periphery and robustness

The study of the core–periphery structure of the network is used to identify the densely connected stations where people can choose more than one path to reach the destination in contrast to sparsely connected stations which can cause a major interruption of the service in case of disruptions. The **core of a network** [39] is computed ranking all the nodes in a network and then counting the number of connections (edges) they have with higher ranked nodes. The node with the highest number of *high-level* connections is the **core border**. All the nodes with a higher ranking than the border node along with the border node itself compose the core of the network, the other nodes are the periphery. A big core suggests several different ways to reach the majority of the nodes and accordingly a more robust network.

### 4.2.1. The rich-club phenomenon

To study the robustness of the networks, we analysed the *rich-club phenomenon* [40]. It is characterized when nodes of higher degree are more interconnected than nodes with lower degree. The presence of this phenomenon may indicate several interesting high-level network properties, such as its robustness. More precisely, this behaviour appears when nodes larger than $k$ are more densely connected among themselves than the nodes with degree smaller than $k$ [41]. This is quantified by computing the *rich-club coefficient* across a range of $k$ values, if this value is greater than 1 for some $k$, the network is characterized by the rich-club phenomenon.

The **rich-club coefficient** for a given network $N$ is usually defined for the degree of the nodes, but it can be generalized to other metrics (e.g. the trophic level). We converted the morning peak-hours directed graph to an undirected one to be consistent with the standard rich-club definition. The generalized formula to compute the rich-club coefficient is as follows:

$$\phi(r) = \frac{2E_{>r}}{N_{>r}(N_{>r}-1)},$$ (4.4)

where only nodes with a richness measure (e.g. node degree rank) of at least $r$ are considered, $E$ is the number of edges and $N$ the number of nodes.

## 4.3. Data-driven analysis of the methods and the parameters

### 4.3.1. Resilience and trophic coherence

In this section, we discuss the properties of the basal node selection methods. Our empirical results show that, on the one hand, the node enforcement always produces incoherent networks (even with a large number of stations enforced we have a high $q/\tilde{q}$ ratio ($>$2)). On the other hand, the passenger filtering technique can achieve a stable network and low incoherence level: eliminating links with a passenger in–passenger out ratio greater than 3.5 creates a network with a $q/\tilde{q}$ of 0.6 (0.4 with passenger in–passenger out ratio $>$8). The latter method is therefore selected, not only because it makes intuitive sense (e.g. a small number of counter-flow passengers is regarded as noise that is filtered out), but also because it creates a network with clear trophic levels that match our qualitative knowledge of how passengers travel.

It is crucial for a reliable analysis to select filter values that are reasonably representative of some underlying data structure. Here, we look for the minimum filter value (so we do not remove too much data), such that the measure of interest (e.g. coherence or core size) is invariant to further filter value changes. Referring to figure 11, for incoherence, we can see that the passenger flow filter method is reasonably stable compared to the enforced basal node method. But, for trophic core size, we can see that a relatively larger value of passenger flow filtering is required. For degree core, we can see that any reasonable filtering or basal node enforcement produces a reliable answer.

As for the overall rail network infrastructure, the rail network in morning peak-hours of each separated company may not have basal nodes, thus to compute its trophic incoherence we firstly

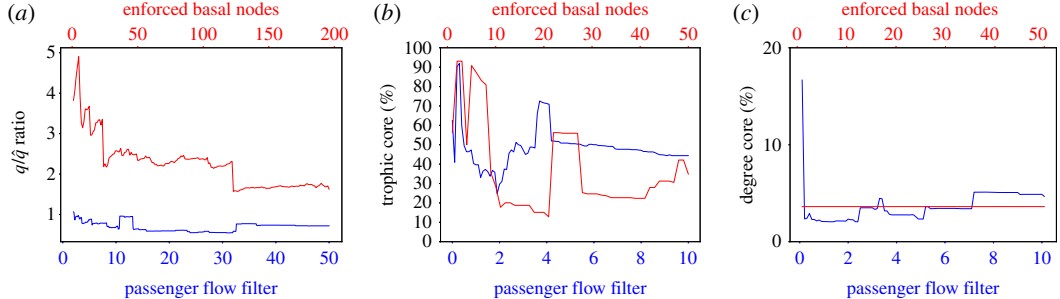

**Figure 11.** Behaviour of the three major resilience and robustness measures used in this work. The basal nodes are selected with several parameter values of the two techniques proposed: the *basal nodes enforcement* parameter is the number of nodes (red) and the *flows filtering* parameter is the filter threshold. (*a*) Incoherence of the network. (*b*) Trophic core – periphery ratio. (*c*) Degree core – periphery ratio.

identified them using one of the techniques described. The *flows filtering* methodology that has proved to be a better technique for our work is used to identify the basal nodes and compute the incoherence parameter $q$ for each company network. To obtain more homogeneous results, the flow filter is applied to all the networks, even if some basal nodes are already existing.

According to the previous results, *flow filtering threshold T* between 1 and 4 provides the best conditions to study the network, filtering the edges that do not represent the studied behaviour (small counter-flows) without modifying significantly the network structure (higher thresholds may also remove interesting flows).

### 4.3.2. Robustness and core – periphery

The study of the core–periphery structure of the network is used to identify the densely connected stations where people can choose more than one path to reach the destination in contrast to sparsely connected stations which can cause a major interruption of the service in case of disruptions. We investigate if a higher percentage of nodes belonging to the core is related to better performances.

Two metrics are used to evaluate the core–periphery of the network. The first metric is relative to the classic definition of network core and rich-club phenomenon: the node degree. The second metric is specific of the studied rail network: as shown previously, in the morning peak-hour rail network, most people travel from the *low trophic level* loosely connected periphery stations to the *high trophic level* well-connected core stations. For this reason, in order to identify the core of the morning peak-hour network, the second metric chosen to rank the stations (nodes) is their trophic level, computed using both the basal node enforcement technique and the passenger flow filter technique.

In this section, we provide a first analysis of the robustness of the morning peak-hour network counting the percentage of nodes that belong to the core of the network, using the degree and trophic level metrics. We will refer to the **degree core** as the core of the network computed ranking the nodes according to their degree, we will refer to the **trophic core** as the core of the network computed ranking the nodes according to their trophic level.

A comparison of the two approaches is shown in figure 11*b,c*. Our results show that the trophic core is much bigger than the degree core. The former core is always between 20 and 90%, the latter is around 5% of the overall nodes. Generally, the passenger flow filter technique generates a network with a bigger core than the enforced basal node network. Our experimental results suggest that while the network is generally not well connected with the higher degree stations (the centre of the network), it is well connected with the stations with a higher trophic level (the morning destination of the passengers).

### 4.3.3. Which basal nodes identification method?

In the London urban rail network during the morning peak-hours, there are no natural basal nodes, so we provided two techniques to artificially select them. Generally speaking, the *Enforcing basal nodes technique* does not modify the structure of the network and can be a good technique when all the edges are important or the graph is unweighted. On the contrary, in scenarios where the difference of the edge weights is significant or the focus is on a certain kind of network behaviour, it may be worth using other approaches. In our study, where the focus is on the morning peak-hour passenger flows, the more reliable

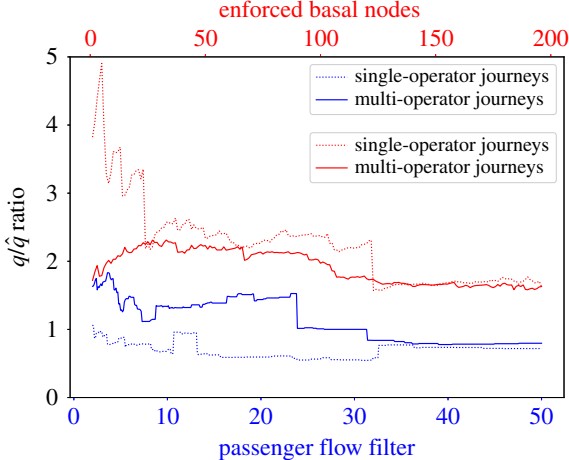

**Figure 12.** Trophic incoherence of the multi-operator rail network (solid lines) compared with the trophic incoherence of the single-operator rail network (dotted lines). The basal nodes are selected using two different methods: the basal nodes enforcement (red) and the passenger flow filtering (blue).

approach consists in using the **flow filtering** method, with a threshold on the *passenger flows* that removes the small counter-flow (e.g. people that live in the centre and work in the suburbs) in order to evidence the mass commute that causes the major stress on the network. With the latter approach and a fair passenger filter threshold, we obtain a network with a trophic coherence similar to a null model. This shows that the rail network during morning peak-hours has a coherence similar to the random expectation, not as incoherent as suspected by enforcing the basal nodes (which does not remove the irrelevant edges).

## 4.4. Multi-operator network

In this paper, we discussed different techniques to measure resilience and robustness of the rail networks during the morning peak-hours, when the network is under stress. We discovered that the trophic incoherence of the network, a measure of the network resilience, is positively correlated with the two main rail performance measures, PPM and CaSL: to higher incoherence is associated a higher probability of delays and disruptions. We computed the trophic incoherence of the rail network of the different operators and the one of the networks obtained merging the different networks in a *single-operator multiplexed rail network* (all the journeys of the different rail operators are represented together in a single graph).

A significant number of commuters reach London in the morning using more than one rail service to complete their journey. For this reason, in this section we complete our study analysing the overall rail network that includes also those journeys. This *multi-operator network* is created merging all the journeys that include a single operator (the previously analysed *single-operator multiplexed rail network*) to the journeys of the commuters that reach their place of work using two or more different rail operators. The multi-operator journeys are computed exploiting the same data sources already discussed in the paper, considering the combination of different operator trains that provides the shortest travel time.

In figure 12, the trophic incoherence (computed using enforced basal nodes and passenger flow filter methods) of the single and multiple operator networks are compared. The results show that the trophic incoherence $(q/\tilde{q})$ of the multi-operator network is similar to the single-operator network one. We can thus state that adding multi-operator journeys does not increase or reduce significantly the overall incoherence of the rail network. Finally, as in the single-operator network (figure 11*a*), the passenger flow filter method is more stable compared to enforced basal node method.

Data accessibility. Data available from the Dryad Digital Repository: https://dx.doi.org/10.5061/dryad.6s76rp7 [42].
Authors' contributions. W.G., L.V., S.Ja., S.Jo. and A.W. planned the experiments. A.P. and G.M. conducted the main numerical analysis. A.A. sourced the data and assisted in the analysis. A.P. and W.G. wrote the paper. All authors helped to analyse the findings and proof-read the paper.
Competing interests. We declare we have no competing interests.
Funding. The authors (W.G. and L.V.) acknowledge funding from EPSRC Engineering Complexity Resilience Network Plus (EP/N010019/1). The authors (W.G., S.Jo. and A.A.) acknowledge funding from EPSRC Centre for Doctoral Training in Urban Science and Progress (EP/L016400/1). The author (G.M.) acknowledges funding from EPSRC & MRC Centre for Doctoral Training in Mathematics for Real World Systems (EP/L015374/1). The authors (A.P.,

A.W. and W.G.) acknowledge funding from the Lloyd's Register Foundation's Programme for Data-Centric Engineering at The Alan Turing Institute.

Acknowledgements. The authors acknowledge Office for National Statistics and the Office for Rail and Road for providing the data and TransportApi for journey planning.

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
