## [Reviewer comments · Royal Society Open Science]

Review History

RSOS-181301.R0 (Original submission)

Review form: Reviewer 1 (Christos Ellinas)

Is the manuscript scientifically sound in its present form?

Yes

Are the interpretations and conclusions justified by the results?

Yes

Is the language acceptable?

Yes

Is it clear how to access all supporting data?

Yes

Do you have any ethical concerns with this paper?

No

Have you any concerns about statistical analyses in this paper?

No

Recommendation?

Major revision is needed (please make suggestions in comments)

Comments to the Author(s)

The authors present an interesting study of the London Urban Rail Network, with a particular focus on linking its performance to certain topological features, which themselves are associated with robustness and/or resilience. Overall, the analysis is sound, and the paper adequately written. However, there are numerous aspects which need to be reworked and polished before this paper can be published. Therefore, I suggest Accept with Major Revisions on this instance. My comments are included in the attached pdf (Appendix A).

Review form: Reviewer 2

Is the manuscript scientifically sound in its present form?

Yes

Are the interpretations and conclusions justified by the results?

Yes

Is the language acceptable?

Yes

Is it clear how to access all supporting data?

Yes

Do you have any ethical concerns with this paper?

No

Have you any concerns about statistical analyses in this paper?

No

Recommendation?

Major revision is needed (please make suggestions in comments)

Comments to the Author(s)

There is probably good work here, but it is hard to assess in the current form, because of lack of clarity in the text in many places.

Abstract: it is not clear what "our" refers to in the first sentence. Is this to be a UK-only study? And what does "network dimensions" mean? Did the authors mean something like "network structure"?

Introduction: needs expansion to properly explain what the aim of the work is. The phrase "in order to faithfully replicate behaviour" is unclear - behaviour of what?

Figure 1: this is completely cryptic to me. What do the graphs mean? The caption "The rail network in morning peak-hours are analysed" should read "... is analysed".

Figure 2: Chiltern Railways and Southeastern are drawn in the wrong places, and there is a typo "Chiltren".

Figure 9: two variables are labelled "%", but the scale only goes to 1.

Footnote superscripts should come after fullstops, not before.

Eqn (4.1) - textual superscripts to mathematical symbols should be in text mode - thus k_i^{in} .

"80 Km" should be "80 km". "Km" would mean Kelvin-metre!

Page 19 line 38 should not be indented. Line 54 should have a fullstop.

"percentage of train either cancelled ..." should be "percentage of trains either cancelled ...".

The heading "(iv) Random Node Removal Comparison" has too many adjectival nouns. Better would be "(iv) Comparison of removal of random nodes".

The use of the slash symbol should be avoided because of its ambiguity - "removal/attack" (not "removal / attack") should be "removal or attack", if that is what is meant. "and / or" should be "and/or".

All graphs are of poor graphic quality, with thin lines and weak fonts. These should be redone with publication-quality software, e.g. matplotlib.

Bibliography: This is all very sloppy, with character encoding errors like "CaÃ'sizares", and initials wrongly formatted, as in "Rodrigue JP" which should be "Rodrigue, J. P.".

Decision letter (RSOS-181301.R0)

03-Sep-2018

Dear Dr Pagani,

The editors assigned to your paper ("Resilience or Robustness: Identifying Topological Vulnerabilities in Rail Networks") have now received comments from reviewers. We would like you to revise your paper in accordance with the referee and Associate Editor suggestions which can be found below (not including confidential reports to the Editor). Please note this decision does not guarantee eventual acceptance.

Please submit a copy of your revised paper before 26-Sep-2018. Please note that the revision deadline will expire at 00.00am on this date. If we do not hear from you within this time then it will be assumed that the paper has been withdrawn. In exceptional circumstances, extensions may be possible if agreed with the Editorial Office in advance. We do not allow multiple rounds of revision so we urge you to make every effort to fully address all of the comments at this stage.

If deemed necessary by the Editors, your manuscript will be sent back to one or more of the original reviewers for assessment. If the original reviewers are not available, we may invite new reviewers.

- Data accessibility

If you wish to submit your supporting data or code to Dryad (<http://datadryad.org/>), or modify your current submission to dryad, please use the following link:
<http://datadryad.org/submit?journalID=RSOS&manu=RSOS-181301>

- Competing interests

- Authors' contributions

- Acknowledgements

- Funding statement

Please note that Royal Society Open Science charge article processing charges for all new submissions that are accepted for publication. Charges will also apply to papers transferred to Royal Society Open Science from other Royal Society Publishing journals, as well as papers submitted as part of our collaboration with the Royal Society of Chemistry (<http://rsos.royalsocietypublishing.org/chemistry>). If your manuscript is newly submitted and subsequently accepted for publication, you will be asked to pay the article processing charge, unless you request a waiver and this is approved by Royal Society Publishing. You can find out more about the charges at <http://rsos.royalsocietypublishing.org/page/charges>. Should you have any queries, please contact openscience@royalsociety.org.

on behalf of Dr Robert MacKay (Associate Editor) and Prof. Mark Chaplain (Subject Editor)
openscience@royalsociety.org

Associate Editor's comments (Dr Robert MacKay):

Both reviewers raise substantial issues. We'd be pleased to receive a major revision of the paper taking their comments into account.

Comments to Author:

Reviewers' Comments to Author:

Reviewer: 1

Comments to the Author(s)

The authors present an interesting study of the London Urban Rail Network, with a particular focus on linking its performance to certain topological features, which themselves are associated with robustness and/or resilience. Overall, the analysis is sound, and the paper adequately written. However, there are numerous aspects which need to be reworked and polished before

this paper can be published. Therefore, I suggest Accept with Major Revisions on this instance. My comments are included in the attached pdf.

Reviewer: 2

Comments to the Author(s)

There is probably good work here, but it is hard to assess in the current form, because of lack of clarity in the text in many places.

Abstract: it is not clear what "our" refers to in the first sentence. Is this to be a UK-only study? And what does "network dimensions" mean? Did the authors mean something like "network structure"?

Introduction: needs expansion to properly explain what the aim of the work is. The phrase "in order to faithfully replicate behaviour" is unclear - behaviour of what?

Figure 1: this is completely cryptic to me. What do the graphs mean? The caption "The rail network in morning peak-hours are analysed" should read "... is analysed".

Figure 2: Chiltern Railways and Southeastern are drawn in the wrong places, and there is a typo "Chiltren".

Figure 9: two variables are labelled "%", but the scale only goes to 1.

Footnote superscripts should come after fullstops, not before.

Eqn (4.1) - textual superscripts to mathematical symbols should be in text mode - thus k_i^{in} .

"80 Km" should be "80 km". "Km" would mean Kelvin-metre!

Page 19 line 38 should not be indented. Line 54 should have a fullstop.

"percentage of train either cancelled ..." should be "percentage of trains either cancelled ...".

The heading "(iv) Random Node Removal Comparison" has too many adjectival nouns. Better would be "(iv) Comparison of removal of random nodes".

The use of the slash symbol should be avoided because of its ambiguity - "removal/attack" (not "removal / attack") should be "removal or attack", if that is what is meant. "and / or" should be "and/or".

All graphs are of poor graphic quality, with thin lines and weak fonts. These should be redone with publication-quality software, e.g. matplotlib.

Bibliography: This is all very sloppy, with character encoding errors like "Cañ´sazes", and initials wrongly formatted, as in "Rodrigue JP" which should be "Rodrigue, J. P.".

Author's Response to Decision Letter for (RSOS-181301.R0)

See Appendix B.

RSOS-181301.R1 (Revision)

Review form: Reviewer 1 (Christos Ellinas)

Is the manuscript scientifically sound in its present form?

Yes

Are the interpretations and conclusions justified by the results?

Yes

Is the language acceptable?

Yes

Is it clear how to access all supporting data?

Yes

Do you have any ethical concerns with this paper?

No

Have you any concerns about statistical analyses in this paper?

No

Recommendation?

Accept as is

Comments to the Author(s)

I would like to thank the Authors for the extensive revisions to the paper, which have strengthened the paper.

A few last minor points (no need to resubmit):

- "The UK rail network transport more"  transports
- Numbers are sometimes conveyed numerical and other times with words (e.g., "[...] measure q/\tilde{q} has a value close to one when a network has a trophic coherence [...] it has a value lower than 1"). Make sure this is unified throughout

Review form: Reviewer 2

Is the manuscript scientifically sound in its present form?

Yes

Are the interpretations and conclusions justified by the results?

Yes

Is the language acceptable?

Yes

Is it clear how to access all supporting data?

Yes

Do you have any ethical concerns with this paper?

No

Have you any concerns about statistical analyses in this paper?

No

Recommendation?

Accept with minor revision (please list in comments)

Comments to the Author(s)

Despite the author's claim to have ensured that "textual superscripts to mathematical symbols should be in text mode", this is still not the case with "out" on page 25. Apart from this I recommend publication in this form.

Decision letter (RSOS-181301.R1)

17-Dec-2018

Dear Dr Pagani:

On behalf of the Editors, I am pleased to inform you that your Manuscript RSOS-181301.R1 entitled "Resilience or Robustness: Identifying Topological Vulnerabilities in Rail Networks" has been accepted for publication in Royal Society Open Science subject to minor revision in accordance with the referee suggestions. Please find the referees' comments at the end of this email.

The reviewers and Subject Editor have recommended publication, but also suggest some minor revisions to your manuscript. Therefore, I invite you to respond to the comments and revise your manuscript.

- Ethics statement

- Data accessibility

<http://datadryad.org/submit?journalID=RSOS&manu=RSOS-181301.R1>

- **Competing interests**

- **Authors' contributions**

- **Acknowledgements**

- **Funding statement**

Because the schedule for publication is very tight, it is a condition of publication that you submit the revised version of your manuscript before 26-Dec-2018. Please note that the revision deadline will expire at 00.00am on this date. If you do not think you will be able to meet this date please let me know immediately.

on behalf of Dr Robert MacKay (Associate Editor) and Mark Chaplain (Subject Editor)
openscience@royalsociety.org

Associate Editor Comments to Author (Dr Robert MacKay):

Associate Editor: 1

Comments to the Author:

I recommend acceptance for publication subject to the (very) minor revisions suggested by the two reviewers.

Reviewer comments to Author:

Reviewer: 1

Comments to the Author(s)

I would like to thank the Authors for the extensive revisions to the paper, which have strengthened the paper.

A few last minor points (no need to resubmit):

- "The UK rail network transport more"  transports

- Numbers are sometimes conveyed numerical and other times with words (e.g., "[...] measure q/\backslash tilde q has a value close to one when a network has a trophic coherence [...] it has a value lower than 1"). Make sure this is unified throughout

Reviewer: 2

Comments to the Author(s)

Despite the author's claim to have ensured that "textual superscripts to mathematical symbols should be in text mode", this is still not the case with "out" on page 25. Apart from this I recommend publication in this form.

Author's Response to Decision Letter for (RSOS-181301.R1)

See Appendix C.

Decision letter (RSOS-181301.R2)

03-Jan-2019

Dear Dr Pagani,

I am pleased to inform you that your manuscript entitled "Resilience or Robustness: Identifying Topological Vulnerabilities in Rail Networks" is now accepted for publication in Royal Society Open Science.

on behalf of Dr Robert MacKay (Associate Editor) and Mark Chaplain (Subject Editor)
openscience@royalsociety.org

Appendix A

Review for “Resilience or Robustness: Identifying Topological Vulnerabilities in Rail Networks” (RSOS-181301)

Reviewer: Dr. Christos Ellinas; Engineering Mathematics, University of Bristol (UK)

Overall

The authors present an interesting study of the London Urban Rail Network, with a particular focus on linking its performance to certain topological features, which themselves are associated with robustness and/or resilience. Overall, the analysis is sound, and the paper adequately written. However, there are numerous aspects which need to be reworked and polished before this paper can be published. Therefore, I suggest Accept with Major Revisions on this instance. My comments are included below:

Structure

Lack of a Literature Review. As it stands, the paper contains no explicit ‘Background’ section – nor any associated Lit Rev – which challenges the ability of the Reader to embrace the context and appreciate this paper’s contribution.

Writing

Paper needs tighter writing. Writing needs to be improved, particularly focusing on: (i) inconsistent sentences, and (ii) inappropriate sentencings. As an example of point (i), the sentence in p.1, line 8-10 makes two related but distinct points which are not well aligned: “Transportation networks are challenging to model” refers to a general modelling challenge related to the system of interest; “large-scale agent based simulation models need to include passenger behaviour [...]” refers to the specific challenges of using agent based modelling for this situation. Despite both being correct points, they are not linked in any consistent manner within this sentence, leaving the Reader unsure on what its purpose is. A second example can be found in p.3, line 26-28, which states that “The challenge with defining trophic coherence in real data driven networks is that it requires some pre-processing: unlike the already studied networks in other works (e.g., food webs) the London Urban Rail Network in peak-hours does not have predefined basal nodes”. What the Authors seem to want to convey is that this specific dataset requires pre-processing because there is no single root or leaf node(s) (i.e., nodes with in-degree or out-degree of 0, respectively). However, this sentence is rather inconsistent, insinuating that the challenge lies in the fact that this network requires pre-processing, which is false, as all networks require some pre-processing to make them usable/useful. Also note that food webs are indeed, real data-derived [not driven] networks. Another example can be found in p.8, line 44-46. As an example of point (ii), several sentences begin with “On the other hand” (e.g., p.1, line 44) without a prior point being made using the sentence “On one hand”. This sentencing is imprecise and is good practice to be avoided.

Sentences need to be correctly supported. Several points are made without appropriate referencing which undermine the confidence of the Reader on their validity. For example, sentence in p.1, line 35-37 makes a grand statement (e.g., “Vulnerability is a major problem in the study of complex networks”) without proper support i.e., no evidence or references. In conjunction with the lack of a dedicated ‘Background’ section, this sort of statements offer no value to the paper and can undermine the confidence of the Reader to the Authors. A somewhat related example is the sentence in p.1, line 54-55: “However, such approaches depend strongly on assumptions about the details of the dynamics or the number of neighbours required for a node to function”. In this case, the Authors use a classical paper to justify this point [May 1972]. However, it is increasingly hard to

convince the Reader that the use of a +40 year old reference can justify a knowledge gap for a 2018 study. Another relevant example can be found in p.3, line 44-45 which argues that since robustness “is not rigorously defined, proxy measures may yield a more holistic insight”. In this instance the Reader is led into an inconsistent line of argument: Are the authors arguing that because robustness is not rigorously defined, measures that are also not rigorously defined are a better fit? Why would that be the case?

Certain sentences which are central to the thesis of the paper are not appropriately supported. For example, p.3., line 15-16 states that “The rationale is that hierarchical systems have fewer feedback loops and are less likely to suffer from cascade effect”; this sentence is central as it makes the key association of feedback loops being important to cascades. The Authors need to expand further, and provide further support, on the validity of this sentence and why this is true as currently, is not sufficiently convincing. This is important as certain topological features may promote or suppress cascading effects, depending on the precise context (e.g., hubs can promote simple contagion, yet can suppress complex contagions).

Footnotes need to be removed. I have not come across other Open Science papers which utilise footnotes, and seeing that footnotes are not explicitly allowed in the Guidelines, I would suggest their removal.

Analysis framework

Analysis is not truly multiplex. I am not convinced that the network examined here is truly multiplex. Seeing that all links correspond to the same function (i.e., people flow), it seems that the network is simply London's Urban rail network (as stated in Fig 2). This does not undermine the value of the analysis offered by the Authors, and by being clearer on this point can actually strengthen the contribution of the paper. On a related point, the paper is not clear whether the results focus on the entire Urban Network (which seems to be the case) or on the subset networks that correspond to each operator

Data & Results

PPM and CaSL appear to be strongly correlated. Authors should provide a Figure plotting the two quantities (incl. some form of correlation measure) in order to identify possible correlations, which appear to be present.

On a related point, I have a question on the nature of the data: are trains that initially experience minor delays (and therefore accounted for in PPM) and eventually delayed more than 30 min (and therefore accounted for in CaSL) included in both measures? If so, then It would be expected that the PPM and CaSL would be strongly related. If so, then the Authors could potentially just use one of them as a measure of performance, simplifying the analysis and sharpening the take away message.

Plotting approach. I wonder if there is any particular reason of why the Authors have not used box plots when presenting the results, as they could provide a cleaner way of conveying the results, without insinuating that the y-axis quantity is a function of the x-axis quantity (which current plots do).

Define measures before use. Authors need to minimally define the measures before they use them in Figures. For example, q/q^A ratio is use in Figure 3 before introduced or defined. This approach forces the Reader to go back and forth in the paper to identify the meaning of the quantity, which is not good practice.

Methods

Define symbols before their use. Define each symbol before being used. For example, eq. 4.1 uses k_i^{in} , a_j and s_j without defining them. Similarly, certain symbols are not use as they should be in the text e.g., in p.11, line 15, the Authors states “out_degree/in_degree” whilst they have previously introduced the k_i^{in} formulation, which they could easily amend and introduce here.

Sensitivity of flows filtering. It would be very useful for both this work, but also for future work, that build on this approach, to illustrate the sensitivity of the approach as a function of T

Null model. The authors should elaborate with a few sentences on the nature of the null model (i.e., basal ensemble) in Section 4c (i).

Minor

Switch Figure order. I would suggest switching the order of Figure 2 with Figure 1, as the latter is a more appropriate introduction to the dataset, whilst Figure 2 is more methodological.

Figure numbering error. Figure 10-12 and caption for Figure 13 should all be replaced by Figure 10a-10c.

Please proof read the paper to eliminate the few typos present

Appendix B

Dear Royal Society Open Science Editorial Office,

Thank you very much for e-mailing us the review comments regarding our manuscript entitled "Resilience or Robustness: Identifying Topological Vulnerabilities in Rail Networks". We have revised the manuscript and include a point by point response to the reviewers suggestions below. All the changes in the revised version of the manuscript have been highlighted in red so that they can be easily found. Finally, we would like to express our gratitude to the esteemed referees for their suggestions.

Yours sincerely,
The Authors

1. Reviewer 1

(a) Literature Review

Specific Comment: *Lack of a Literature Review. As it stands, the paper contains no explicit "Background" section - nor any associated Lit Rev - which challenges the ability of the Reader to embrace the context and appreciate this paper's contribution.*

Response: We have re-written the Literature Review and Background sections by adding more examples of thematically similar work and better explained our choice for measures of network resilience and robustness. Please see below:

(i) Related Work

UK rail network The UK rail network transport more than 1.7 billion passengers per year, of which 1.1 billion passengers commute in the London area [1]. According to the Office of Rail and Road [2], in the last year in the London area, only 86.9% passenger trains arrived on time and 4.8% of the journeys experienced cancellations or significantly lateness. Often these delays are interrelated and the relationship between cascade effects and network dynamics is not well understood.

In current literature, most of the proposed studies consider natural or man-made disasters, but they do not consider the stress of the network during the peak-hours and how the structure of the network created by the massive flows of people can influence their ability to maintain a good service. For example, several graph based approaches have been proposed to improve the performances by revising the design and maintenance of the rail networks [4,7], but do not consider dynamic passenger flows. Other studies focus on specific extreme scenarios [5,6] or unfavorable conditions [3] that cause disruptions.

As our data shows, under the same external conditions, the major rail companies in and around London reach dramatically different performance levels. In this work we **hypothesize** that this difference can in part be attributed to the peak passenger demand. A coupling relationship between flow and network structure can tease out the indicative measures that correlate strongly with overall performance.

Vulnerability of Transport Networks The concept of vulnerability of transportation network, introduced in the literature by Berdica [8], is generally defined as the susceptibility to disruptions that could cause considerable reductions in network service or the ability to use a particular network link or route at a given time. Many have applied general network science disruption analysis. For example, several studies [11–13] have been conducted for modelling railway vulnerability with promising predictive results. Bababeik et al. [10] recently proposed a mathematical programming model that is able to identify critical links with consideration of supply and demand interactions under different disruption scenarios. Recent work has also used graph properties to infer interaction strengths and use an epidemic spreading model to predict delays in railway networks [22].

(b) Paper needs tighter writing

Response: We have improved the rigour and logic of our proposed method. Please see both the revised Introduction and Methods sections.

(c) Sentences need to be correctly supported.

Specific Comment: Several points are made without appropriate referencing which undermine the confidence of the reader on their validity. For example, sentence in p.1, line 35-37 makes a grand statement (e.g., "Vulnerability is a major problem in the study of complex networks") without proper support i.e., no evidence or references.

Response: We have supported this in the **Related Work** section.

Specific Comment: "However, such approaches depend strongly on assumptions about the details of the dynamics or the number of neighbours required for a node to function". In this case, the Authors use a classical paper to justify this point [May 1972]. However, it is increasingly hard to convince the Reader that the use of a +40 year old reference can justify a knowledge gap for a 2018 study

Response: We have updated our methodological review for resilience and robustness, please see below:

The concepts of resilience and robustness on networks admit various interpretations and definitions [23,26]. A generally accepted definition of stability is for when the system performance returns to a desirable state. For homogeneous linear stability, one might equate resilience with equilibrium points and look at the leading eigenvalue of the Jacobian matrix [24]. When linear stability is not suitable due to complex dynamics, many authors [15–19] have studied system resilience from different perspectives. Some consider the dynamic response (e.g. time to recovery) of the whole system after a specific disruption [19], whilst others use random perturbations to numerically quantify system response [20].

In terms of robustness, a common definition is the number of nodes that must be removed in order for the network to break down is a popular measure of its robustness [25]. Whilst such approaches depend strongly on assumptions about the system, it generally maps well to railway systems [14].

Specific Comment: Another relevant example can be found in p.3, line 44-45 which argues that since robustness "is not rigorously defined, proxy measures may yield a more holistic insight". In this instance the reader is led into an inconsistent line of argument: Are the authors arguing that because robustness is not rigorously defined, measures that are also not rigorously defined are a better fit? Why would that be the case?

Response: Our logic is as follows. As there exist no singular mathematically grounded logic for robustness, a variety of robustness measures are used to compare against real railway performance. As such, we are not arguing that alternative robustness measures are better, but that we are using a variety to establish a wider evidence base.

Specific Comment: Certain sentences which are central to the thesis of the paper are not appropriately supported. For example, p.3., line 15-16 states that "The rationale is that hierarchical systems have fewer feedback loops and are less likely to suffer from cascade effect"; this sentence is central as it makes the key association of feedback loops being important to cascades. The Authors need to expand further, and provide further support, on the validity of this sentence and why this is true as currently, is not sufficiently convincing. This is important as certain topological features may promote or suppress cascading effects, depending on the precise context (e.g., hubs can promote simple contagion, yet can suppress complex contagions).

Response: We substantiate our proposed method more rigorously here. When networks are modeled as a discrete linear time invariant (LTI) system with a defined input and output [21], the dynamic response stability is defined by the location of roots of its transfer function (negative

domain). In such a case, absence of feedback loops ensures stability. Presence of feedback loops will cause non-zero roots and risk instability. When we consider a complex network with $\sim N^2$ input output combinations, the transfer function cannot be defined. As such, we measure the overall network incoherence, which is a compressed figure of merit for how many feedback loops exist.

(d) Footnotes need to be removed

Response: We moved the footnotes to the main text.

(e) Analysis is not truly multiplex.

Specific Comment: I am not convinced that the network examined here is truly multiplex. Seeing that all links correspond to the same function (i.e., people flow), it seems that the network is simply London's Urban rail network (as stated in Fig 2). This does not undermine the value of the analysis offered by the Authors, and by being clearer on this point can actually strengthen the contribution of the paper. On a related point, the paper is not clear whether the results focus on the entire Urban Network (which seems to be the case) or on the subset networks that correspond to each operator

Response: We consider it a multiplex network because it is the composition of different railway networks with some overlapping nodes (multiplex stations). Multiplex stations of different companies cannot be collapsed together because they reach different points (stations) depending on the network they belong to. On one hand the passenger flows can be merged by adding up the flow volumes in the links shared between companies, this general network is then used to perform further analysis on the overall resilience and robustness and to define some common parameters (e.g., filter flows threshold). On the other hand, the rail networks are studied separately to measure their robustness and resilience and to compare these values with their performances (i.e., PPM and CaSL).

(f) PPM and CaSL appear to be strongly correlated

Specific Comment: Authors should provide a Figure plotting the two quantities (incl. some form of correlation measure) in order to identify possible correlations, which appear to be present. On a related point, I have a question on the nature of the data: are trains that initially experience minor delays (and therefore accounted for in PPM) and eventually delayed more than 30 min (and therefore accounted for in CaSL) included in both measures? If so, then It would be expected that the PPM and CaSL would be strongly related. If so, then the Authors could potentially just use one of them as a measure of performance, simplifying the analysis and sharpening the take away message.

Response: PPM provides the percentage of trains that reach their destination station within 5 minutes. CaSL measures the percentage of trains that are cancelled or significantly in delay (more than 30 minutes). In our paper we define oPPM as the opposite of PPM (oPPM = 100% - PPM) to evidence this possible correlation between PPM and CaSL. Generally speaking, we expect some correlation but we want to discriminate companies that struggle to reach their destination in time but never have major disruptions (high oPPM and low CaSL) from companies that usually perform well but suffer several delays in case of failure (oPPM = CaSL).

4 out of 5 analysed rail companies (see Figure 1) show a strong correlation between these 2 measures while in one case (Great Western Railway) these values are not correlated, possibly meaning that this company often has little delays (low resilience) but generally does not have major disruptions (high robustness).

We used both measures to analyse the performances of the rail networks in order to consider all the possible scenarios and to provide more detailed results for those who want to compare the rail companies performances in other cities.

Figure 1. oPPM vs CaSL correlation

(g) Plotting approach

As recommended by the reviewer, we have re-plotted some results as box-plots, which are in Section 3.

(h) Define measures before use.

Specific Comment: Authors need to minimally define the measures before they use them in Figures. For example, q/\bar{q} ratio is used in Figure 3 before introduced or defined. This approach forces the Reader to go back and forth in the paper to identify the meaning of the quantity, which is not good practice.

Response: We added this paragraph at the beginning of Section 2.a to introduce the measures: The degree to which empirical networks are coherent (or incoherent) can be investigated by comparison with a null model. We use the basal ensemble expectation \bar{q} as a null model to compare the incoherence parameter of our rail networks (more details and the computation of the basal ensemble expectation are provided in Section Methods,). The trophic incoherence measure q/\bar{q} has a value close to one when a network has a trophic coherence similar to a random expectation, it has a value lower than 1 when the network is coherent while it has a value greater than 1 when the network is incoherent.

We replaced this sentence in section 2.a:

The comparison between the trophic incoherence measure q/\bar{q} and the performance measures oPPM and CaSL is shown in Figure 6.

with:

The trophic incoherence measure q/\bar{q} compared with oPPM and CaSL is shown in details in Figure 6: more coherent networks (low q/\bar{q}) are generally associated with lower delays (oPPM) and cancellations (CaSL).

(i) Define symbols before their use

Specific Comment: Define each symbol before being used. For example, eq. 4.1 uses k_i^{in} , a_{ij} and s_j without defining them. Similarly, certain symbols are not used as they should be in the text e.g., in p.11, line 15, the Authors states *out_degree/in_degree* whilst they have previously introduced the k_i^{in} formulation, which they could easily amend and introduce here.

Response: Below eq 4.1 we clarified the meaning of the symbols used:

1st line trophic level of a node i , called s_i , is defined as...

Equation 4.1

where a_{ij} is the adjacency matrix of the graph and $k_i^{in} = \sum_j a_{ij}$ is the number of in-neighbours (in degree) of the node i . Basal nodes $k_i^{in} = 0$ have trophic level $s_i = 1$ by convention.

A partial rewriting of the entire section has been performed as well.

In section 4.4.iii, the definition:

The largest strongly connected component in the network is identified and, given a number of basal nodes, the nodes with the lower out_degree/in_degree ratio are enforced to be basal nodes (thus with trophic level 1).

is replaced with the more precise and coherent:

This technique enforces a predefined number EN of nodes to be basal nodes (their trophic level is imposed 1). The nodes to be enforced are selected according to their similarity to real basal nodes, namely the nodes with the lowest ratio between incoming and outgoing edges. More formally, the k^{out}/k^{in} ratio is computed for all the nodes, then the trophic level of the EN nodes with the lower ratio is enforced to 1 ($s_i = 1$). If parts of the network are not connected to basal nodes, only the largest strongly connected component considered.

(j) Sensitivity of flows filtering

Specific Comment: *It would be very useful for both this work, but also for future work, that build on this approach, to illustrate the sensitivity of the approach as a function of T*

Response: In Figure 2 we show the behavior of the three measures used in this work for different values of the threshold T (flows filtering approach) and of enforced nodes EN (basal nodes enforcement approach).

Figure 2. Behaviour of the three major resilience and robustness measures used in this work. The basal nodes are selected with several parameter values of the two techniques proposed: the *Basal nodes enforcement* parameter is the number of nodes (red) and the *Flows filtering* parameter is the filter threshold.

It is crucial for a reliable analysis to select filter values that are reasonably representative of some underlying data structure. Here, we look for the minimum filter value (so we do not remove too much data), such that the measure of interest (e.g. coherence or core size) is invariant to further filter value changes. Referring to Figure 2. For incoherence, we can see that Passenger flow filter method is reasonably stable compared to enforced basal node method. Whereas, for trophic core size, we can see that a relatively larger value of passenger flow filtering is required. For degree core, we can see that any reasonable filtering or basal node enforcement produces a reliable answer.

(k) Null model

Specific Comment: The authors should elaborate with a few sentences on the nature of the null model (i.e., basal ensemble) in Section 4c (i).

Response: In our work we compared the trophic incoherence of the rail networks with a null model, the basal ensemble. The **basal ensemble expectation** \tilde{q} can be considered a good approximation for finite random networks [27].

The basal ensemble expectation for the incoherence parameter is [27]:

$$\tilde{q} = \sqrt{\frac{L}{L_B} - 1} \quad (1.1)$$

where:

L = number of edges in the network

L_b = number of edges connected to basal nodes

The ratio q/\tilde{q} is used to analyse the coherence of the network: a value close to one shows a network with a trophic coherence similar to a random expectation. Values lower than 1 reveal coherent networks, while values greater than 1 incoherent ones.

For example, Johnson and Jones [27] found that food webs are significantly coherent ($q/\tilde{q} = 0.44 \pm 0.17$), metabolic networks are significantly incoherent ($q/\tilde{q} = 1.81 \pm 0.11$) and gene regulatory networks are close to the random expectation ($q/\tilde{q} = 0.99 \pm 0.05$).

We provide this details in Section 4.a.i.

(l) Switch figure order

Specific Comment: I would suggest switching the order of Figure 2 with Figure 1, as the latter is a more appropriate introduction to the dataset, whilst Figure 2 is more methodological.

Response: We redrew figure 1 to provide a more general explanation of our work and to provide a synthetic description of the approach used in the paper.

(m) Figure numbering error

Response: Fixed. There was an incompatibility with a latex package.

2. Reviewer 2

Response: We have to correct some typos, and revise some phrases:

Specific Comment: Abstract - it is not clear what "our" refers to in the first sentence. Is this to be a UK-only study? And what does "network dimensions" mean? Did the authors mean something like "network structure"?

Response: We corrected the sentence.

Specific Comment: Introduction - needs expansion to properly explain what the aim of the work is. The phrase "in order to faithfully replicate behaviour" is unclear - behaviour of what?

Response: We improved the introduction and provided a more detailed description of the aim of the work:

Cascade delays and cancellations on rail transport can cause devastating economic damage and dent consumer satisfaction. Existing knowledge focus either on improving operational practices or consider pure topological analysis. We advance this, by considering both real passenger travel flows and the network topology together. This creates a stronger understanding of its dynamic vulnerability and resilience.

Specific Comment: Figure 1 - this is completely cryptic to me. What do the graphs mean? The caption "The rail network in morning peak-hours are analysed" should read "... is analysed".

Response: We redrew Figure 1 in order to provide a better explanation of the proposed work, please see Figure 3 in this document.

Specific Comment: Figure 2 - Chiltern Railways and Southeastern are drawn in the wrong places, and there is a typo "Chiltren".

Response: The figure has been corrected, please see Figure 4 in this document.

Specific Comment: Figure 9: two variables are labelled "%", but the scale only goes to 1.

Response: it is the percentage of nodes that belongs to the core, while the scale refers to the correlation index. We redrew the figure to better clarify this point.

Specific Comment: Footnote superscripts should come after fullstops, not before.

Response: We removed the footnotes in order to accomplish the Royal Society Open Science publication standards.

Specific Comment: Eqn (4.1) - textual superscripts to mathematical symbols should be in text mode - thus k_i^{in} .

Response: Thanks, we corrected this.

Specific Comment: "80 Km" should be "80 km". "Km" would mean Kelvin-metre!

Response: Thanks, we corrected this.

Specific Comment: Page 19 line 38 should not be indented. Line 54 should have a fullstop.

Response: Thanks, we corrected this (page 10).

Specific Comment: "percentage of train either cancelled ..." should be "percentage of trains either cancelled ...".

Response: Thanks, we corrected this.

Specific Comment: The heading "(iv) Random Node Removal Comparison" has too many adjectival nouns. Better would be "(iv) Comparison of removal of random nodes".

Response: Thanks, we corrected this.

Specific Comment: The use of the slash symbol should be avoided because of its ambiguity - "removal/attack" (not "removal / attack") should be "removal or attack", if that is what is meant. "and / or" should be "and/or".

Response: Thanks, we corrected this.

Specific Comment: All graphs are of poor graphic quality, with thin lines and weak fonts. These should be redone with publication-quality software, e.g. matplotlib.

Response: The graphs have been redone, please see Section 3 of this document.

Specific Comment: Finally, review the bibliography style.

Response: It has been reviewed according to the Royal Society Open Science format.

3. New plots

Figure 3. New Figure 1.

Figure 4. Directed graph of passenger flows during a morning peak-hours.

Figure 5. replaced with: *please see figure on the right-hand side*

Figure 7. oPPM and CaSL compared with the trophic incoherence measure of each rail company.

Figure 6. replaced with: *please see figure on the right-hand side*

Figure 8. oPPM and CaSL compared with Rich-Club coefficient of each rail company.

Figure 9. replaced with: please see figure on the right-hand side

Figure 10. oPPM and CaSL compared with degree core of each rail company.

Figure 11. replaced with: please see figure on the right-hand side

Figure 12. oPPM and CaSL compared with the trophic core of each rail company.

Figure 13. replaced with: please see figure on the right-hand side

Figure 14. oPPM and CaSL compared with the robustness to attacks.

Figure 15. Pearson Correlation Coefficient between different measures and indicators. Incoherence parameter / basal ensemble (q/\bar{q}), incoherence parameter (q), size of degree core (Size Degree-Core), size of trophic core (Size Trophic-Core), rich-club phenomenon (Rich-club), Robustness to attacks (Attacks), 100% - Public Performance Measure (oPPM), Cancellations and Significant Lateness (CaSL), number of employees, number of stations, number of trains, number of passengers and multiplex stations (stations that connect different companies).

References

1. Passenger rail usage. Office of Rail and Road. Retrieved 10 September 2018. See <http://dataportal.orr.gov.uk/browse/reports/12>.
2. Passenger and freight rail performance. Office of Rail and Road. Retrieved 10 September 2018. See <http://dataportal.orr.gov.uk/browse/reports/3>.
3. Norrbin P, Lin J, Parida A. 2016. Infrastructure Robustness for Railway Systems. *International Journal of Performability Engineering*, Vol. 12, No.3.
4. European Commission, Horizon 2020. 2015. *Smart, Green and Integrated Transport, Work Programme 2016-2017. Brussels, European Commission*.
5. BS EN 50126-2:2017. 1999. Railway Specifications – The Specification and Demonstration of Reliability, Availability, Maintainability and Safety (RAMS). Systems Approach to Safety. *European Committee for Electrotechnical Standardization*
6. Fatouche R, Miller-Hooks E. 2014. Measuring the Performance of Transportation Infrastructure Systems in Disasters: A Comprehensive Review. *Journal of Infrastructure Systems*.
7. Al-Douri YK, Tretten P, Karim R. 2016. Improvement of railway performance: a study of Swedish railway infrastructure. *Journal of Modern Transportation*.
8. Berdica K. 2002. An introduction to road vulnerability: What has been done, is done and should be done. *Transp. Policy*, 117-127.
9. Wang Z, Chan AP, Yuan J, Xia B, Skitmore M, Li Q. 2014. Recent advances in modeling the vulnerability of transportation networks. *Journal of Infrastructure Systems* 21.
10. Bababeik M, Khademi N, Chen A, Nasir MM. 2017. Vulnerability Analysis of Railway Networks in Case of Multi-Link Blockage. *Transportation Research Procedia* Volume 22, Pages 275-284.
11. Gedik R, Medal H, Rainwater C, Pohl EA, Mason SJ. 2014. Vulnerability assessment and re-routing of freight trains under disruptions: A coal supply chain network application. *Transportation Research Part E: Logistics and Transportation Review* 71, 45-57.
12. Khaled AA, Jin M, Clarke DB, Hoque MA. 2015. Train design and routing optimization for evaluating criticality of freight railroad infrastructures. *Transportation Research Part B: Methodological* 71, 71-84.
13. Zhang Z, Li X, Li H. 2015. A quantitative approach for assessing the critical nodal and linear elements of a railway infrastructure. *International Journal of Critical Infrastructure Protection* 8, 3-15.
14. Dehghani MS, Flintsch G, McNeil S. 2014. Impact of Road Conditions and Disruption Uncertainties on Network Vulnerability. *Journal of Infrastructure Systems* 20.
15. Reggiani A, Graaff T, Nijkamp P. 2002. Resilience: an evolutionary approach to spatial economic systems. *Networks Spatial Econ.*, pp. 211-229.
16. Bruneau M, Chang SE, Eguchi RT, Lee GC, O'Rourke TD, Reinhorn AM, Shinouzuka M, Tierney K, Wallace WA. 2003. A framework to quantitatively assess and enhance the seismic resilience of communities Earthquake Spectra, pp. 733-752.
17. Chang SE, Shinouzuka M. 2004, Measuring improvements in the disaster resilience of communities Earthquake Spectra, pp. 739-755.
18. Rose A. 2007. Economic resilience to natural and man-made disasters: multidisciplinary origins and contextual dimensions, *Environ. Hazards*, pp. 383-395.
19. Hallegatte S. 2014. Economic Resilience: Definition and Measurement. Policy Research Working Paper, WPS 6952, The World Bank Group.
20. D'Lima M, Medda F. 2015. A new measure of resilience: An application to the London Underground, *Transportation Research Part A: Policy and Practice*, Volume 81, 2015, Pages 35-46.
21. Proakis J, Manolakis D. 2006. *Digital Signal Processing*, Wiley.
22. Monechi B, Gravino P, Clemente R, Servedio V. 2018. Complex delay dynamics on railway networks from universal laws to realistic modelling, *European Physics Journal: Data Science*, Volume 7.
23. Grimm V, Calabrese J. 2011. *What Is Resilience? A Short Introduction*, volume 11.
24. May RM. 1972. Will a large complex system be stable?, *Nature*, volume 238, pages 413-14. DOI <http://dx.doi.org/10.1038/238413a0>
25. Boldi P, Rosa M, Vigna S. 2013. Robustness of social and web graphs to node removal, *Social Network Analysis and Mining*, volume 3, pages 829-842. (Url <https://doi.org/10.1007/s13278-013-0096-x>, DOI 10.1007/s13278-013-0096-x.)

26. Klau GW, Weiskircher R. 2005. *Robustness and Resilience*, pages 417–437 (Springer Berlin Heidelberg). (Url https://doi.org/10.1007/978-3-540-31955-9_15.)
27. Johnson S, Jones NS. 2017. Looplessness in networks is linked to trophic coherence, *Proceedings of the National Academy of Sciences*, volume 114, pages 5618–5623. (Url <http://www.pnas.org/content/114/22/5618>, DOI 10.1073/pnas.1613786114.)

Appendix C

Dear Editors,

we would like to thank you for accepting our manuscript RSOS-181301.R1 entitled "Resilience or Robustness: Identifying Topological Vulnerabilities in Rail Networks" for publication in Royal Society Open Science.

We revised the paper following the suggestions of the two reviewers. We also added robust data testing and an operator focused case study in the Supplementary Material.

Best regards,

The Authors